# An infalling observer and behind the horizon cutoff

**Amin Akhavan**[1][⋆] **and Mohsen Alishahiha**[2][†]

**1** Young Researchers Club, Central Tehran Branch, Islamic Azad University, Tehran, Iran
**2** School of physics, Institute for Research in Fundamental Sciences (IPM)
P.O. Box 19395-5531, Tehran, Iran

⋆ amin_akhavan@ipm.ir          † alishah@ipm.ir

*Dedicated to Farhad Ardalan on the occasion of his 80$^{th}$ birthday*

## Abstract

Using Papadodimas and Raju construction of operators describing the interior of a black hole, we present a general relation between partition functions of operators describing inside and outside the black hole horizon. In particular for an eternal black hole the partition function of the interior modes may be given in terms those partition functions associated with the modes of left and right exteriors. By making use of this relation we observe that setting a finite UV cutoff will enforce us to have a cutoff behind the horizon whose value is fixed by the UV cutoff. The resultant cutoff is in agreement with what obtained in the context of holographic complexity.

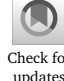
## 1 Introduction

It is of a great interest to understand the physics behind the horizon, though its understanding might require full knowledge of quantum gravity. Nonetheless, AdS/CFT correspondence [1]

has provided a practical tool to explore, at least, certain features of the physics behind the horizon. Working within the framework of AdS/CFT correspondence the problem may be rephrased as how to construct space time from boundary field theory. In this context holographic entanglement entropy [2] was found useful to explore this possibility, though it might not be enough [3].

Holographic complexity, by definition, is a quantity that is sensitive to regions behind the horizon. Indeed, using either proposals for holographic complexity ("complexity=volume" (CV) [3,4] or "complexity=action" (CA) [5,6]) one will have to deal with a portion of space time located behind the horizon. It is either a part of the Einstein-Rosen bridge, or a part of the Wheeler-DeWitt patch (WDW). Therefore one would naturally expect that the holographic complexity may have information of the physics behind the horizon.

In the CA proposal the late time behavior of complexity for an eternal black hole is entirely given by the on shell action evaluated on the intersection of WDW patch with the future interior of the black hole [7]. On the other hand one would expect that the late time behavior of complexity is determined by physical charges, such as energy, which are computed on the boundary of the space time. This fact may indicate a possibility of having a relation between inside and outside the horizon. We note, however, that this relation would result in an apparent puzzle as well. While the physical charges are sensitive to a UV cutoff, the late time behavior of holographic complexity, giving entirely by interior of the black hole, is blind to the UV cutoff.

A remedy to resolve this puzzle was proposed in [8] (see also [9–12]) in which it was argued that a UV cutoff will induce a cutoff behind the horizon whose value is fixed by the UV cutoff.

It is interesting to see if the cutoff behind the horizon could also be seen by other physical quantities. This is, indeed, the aim of this article to explore this possibility. To address this question one needs to look for an object that has a potential to probe physics behind the horizon.

To proceed, we note that, in the context of AdS/CFT correspondence, there are several attempts to construct operators in the dual conformal field describing the interior of a black hole [13,14]. Although there are serious concerns on the state dependence of these constructions [15–17], it is found useful to examine the cutoff behind the horizon in this context. Of course in what follows we will consider an eternal black hole in which the situation is better understood. Indeed, in most of our computations we do not really need the construction of [13] and the bulk description of the operators is enough.

This article is organized as follows. In the next section we will briefly review the construction of "interior operators" proposed in [13]. In section three we shall consider partition functions of the interior and exterior operators where we will also present a relation between them. Then, we will examine this relation when the theory is put at a finite cutoff. Interesting enough, we observe that setting a UV cutoff will automatically induce a cutoff behind the horizon whose value is exactly the one given by the holographic complexity. The last section is devoted to discussions.

## 2 Interior operators

In this section we will review the proposal of [13] to construct conformal field theory operators describing the interior of a black hole. To fix our notation, we will consider an eternal black brane solution of an Einstein gravity with negative cosmological constant. The action and the corresponding solution may be written as follows

$$S_{\text{EH}} = \frac{1}{16\pi G} \int d^{d+2}x \sqrt{g} \left( R + \frac{d(d+1)}{L^2} \right), \tag{1}$$

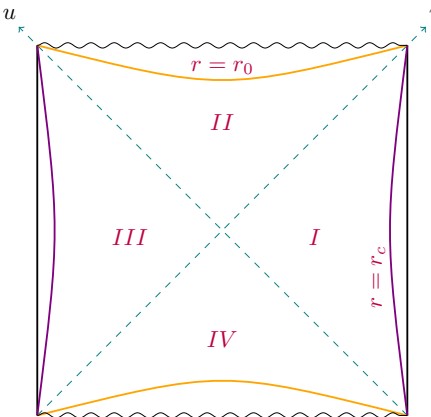

Figure 1: Penrose diagram of an eternal black brane. The UV cutoff is denoted by $r_c$ and the cutoff behind the horizon is denoted by $r_0$.

and

$$ds^2 = \frac{L^2}{r^2}\left(-f(r)dt^2 + \frac{dr^2}{f(r)} + d\vec{x}^2\right), \quad f(r) = 1 - \frac{r^{d+1}}{r_h^{d+1}}. \tag{2}$$

The Penrose digram of the solution is depicted in the Fig. 1. Here $\vec{x}$ are coordinates parametrizing a $d$ dimensional flat space. Note that, there are also certain boundary terms which are needed in order to have a consistent variational principle. It is known that this model provides a gravitational description for a thermofield double state [19].

Let us consider a generalized free field $\mathcal{O}(t, \vec{x})$ in the field theory side whose modes in the momentum space are denoted by $\mathcal{O}_{\omega,\vec{k}}$.[1] Then one may define a CFT (non-local) operator labeled by the AdS radial coordinate $r$ as follows

$$\phi_{\text{CFT}}^{I}(t, \vec{x}, r) = \int_{\omega>0} \frac{d\omega d^d k}{(2\pi)^{d+1}} \left[\mathcal{O}_{\omega,\vec{k}} f_{\omega,\vec{k}}(t, \vec{x}, r) + \text{h.c.}\right]. \tag{3}$$

When the function $f_{\omega,\vec{k}}(t, \vec{x}, r)$ satisfies the Klein-Gordon equation for a free scalar in the back brane solution (2) with normalizability condition near the boundary (and no condition at the horizon), this operator has the same correlators as that of a free-field propagating in the black brane background [13].

This is, indeed, a local bulk operator in region $I$ of the black barne shown in the Fig. 1. Similarly, one may define an operator associated with a free field associated with the region $III$ as follows

$$\phi_{\text{CFT}}^{III}(t, \vec{x}, r) = \int_{\omega>0} \frac{d\omega d^d k}{(2\pi)^{d+1}} \left[\tilde{\mathcal{O}}_{\omega,\vec{k}} \tilde{f}_{\omega,\vec{k}}(t, \vec{x}, r) + \text{h.c.}\right]. \tag{4}$$

Here we have used the "tilde" notation to make a distinction between operators describing regions $I$ and $III$, though both of then are defined in the same dual conformal field theory.

To construct an operator describing the interior of the black brane (the region denoted by $II$ in the Fig. 1) one needs to consider both operators defined by $\mathcal{O}$ and $\tilde{\mathcal{O}}$. More precisely, one has [13]

$$\phi_{\text{CFT}}^{II}(t, \vec{x}, r) = \int_{\omega>0} \frac{d\omega d^d k}{(2\pi)^{d+1}} \left[\mathcal{O}_{\omega,\vec{k}} \, g_{\omega,\vec{k}}^{(1)}(t, \vec{x}, r) + \tilde{\mathcal{O}}_{\omega,\vec{k}} \, g_{\omega,\vec{k}}^{(2)}(t, \vec{x}, r) + \text{h.c.}\right], \tag{5}$$

---

[1] Note that by generalized free field one means that the correlators of the corresponding operators factorize at large $N$ limit [18].

where $g^{(1)}$ and $g^{(2)}$ still satisfy the Klein-Gordon equation for a free scalar in the back brane solution, of course with no boundary condition. One just needs to impose the continuity of the field at the horizon between regions *I* and *II* and also *III* and *II*. For more details of how to construct these operators the reader is refereed to the original paper [13].

## 3 Partition function and cutoff

In this section we would like to study possible information one could get from the physics behind horizon using the operator given in the equation (5) describing the interior of the black hole. Intuitively, from the construction of the operator (5) one may extract following information.

First of all it seems that in order to study the region *II* one needs twice the number of modes as those in region *I*. Secondly, since the operator (5) is a non-local operator in the dual field theory whose non-locality parameter is given by the AdS redial coordinate, imposing any restriction on the non-locality parameter (such as setting a UV cutoff) would restrict the range of the space time accessible to those fields defined behind the horizon.

Clearly these facts make a connection between inside and outside the horizon's physics, reminiscing the phenomena observed in the complexity computations [8]. Indeed, the aim of this section is to understand this connection better.

To explore a possible connection, let us study the partition function of the scalar field we have considered in the previous section. It is worth noting that since we are working within the context of AdS/CFT correspondence in which there is a relation between partition functions of gauge theory and gravity, it what follows when we are computing the partition function we will not make an explicit distinction between these two pictures. Of course from the context, it should be clear whether we are using the partition function in the gauge theory or gravity descriptions.

To proceed, starting from the field theory description, let us consider a *restricted* partition function in which the integration is taken over the fields associated with regions *I* or *III* of the corresponding eternal black hole[2]

$$Z^{(I)} \propto \int \mathcal{D}\phi^{(I)} \, e^{-iS[\phi^{(I)}]}, \qquad Z^{(III)} \propto \int \mathcal{D}\phi^{(III)} \, e^{-i\tilde{S}[\phi^{(III)}]}. \tag{7}$$

By making use of the explicit expressions of the corresponding fields in terms on $\mathcal{O}$ and $\tilde{\mathcal{O}}$ given in the equations (3) and (4) one finds

$$Z^{(I)} \propto \int_{\omega>0} \mathcal{D}\mathcal{O}_{\omega,\vec{k}} \, \mathcal{D}\mathcal{O}_{-\omega,-\vec{k}} \, e^{-iS[\mathcal{O}]}, \qquad Z^{(III)} \propto \int_{\omega>0} \mathcal{D}\tilde{\mathcal{O}}_{\omega,\vec{k}} \, \mathcal{D}\tilde{\mathcal{O}}_{-\omega,-\vec{k}} \, e^{-i\tilde{S}[\tilde{\mathcal{O}}]}. \tag{8}$$

One may find a similar expression for the field describing the regions *II* which, of course, should contain both sets of the operators $\mathcal{O}$ and $\tilde{\mathcal{O}}$

$$Z^{(II)} \propto \int_{\omega>0} \mathcal{D}\mathcal{O}_{\omega,\vec{k}} \, \mathcal{D}\tilde{\mathcal{O}}_{\omega,\vec{k}} \mathcal{D}\mathcal{O}_{-\omega,-\vec{k}} \, \mathcal{D}\tilde{\mathcal{O}}_{-\omega,-\vec{k}} \, e^{-i\bar{S}[\mathcal{O},\tilde{\mathcal{O}}]}. \tag{9}$$

---

[2] A motivation of considering this *restricted* partition function may be understood from the gravitional description of the model. Indeed, starting from partition function in the gravity daul one may approximate it by restricting our path integral over those field configurations satisfying the equations of motion (saddle point approximation)

$$Z \propto \int \mathcal{D}\phi \, e^{-iS[\phi]} \delta(\text{e. o. m.}), \tag{6}$$

where e. o. m. stands for "equations of motion" and therefore the path integral should be performed for those fields given in the form of (3), (4) or (5).

In general, the action appearing in the expression of (9) would be a complicated function of $\mathcal{O}$ and $\tilde{\mathcal{O}}$ and it cannot be decomposed in terms of actions for operators $\mathcal{O}$ and $\tilde{\mathcal{O}}$, separately; $\bar{S}[\mathcal{O}, \tilde{\mathcal{O}}] \neq S[\mathcal{O}] + \tilde{S}[\tilde{\mathcal{O}}]$. It is also important to note that a field in the region $II$, is not a liner combination of those defined outside the horizon. Nonetheless for generalized free fields and taking large $N$ limit in which the corresponding correlation functions of the operators factorize (see [18])

$$\langle \mathcal{O}_1 \cdots \mathcal{O}_n \tilde{\mathcal{O}}_1 \cdots \tilde{\mathcal{O}}_m \rangle = \langle \mathcal{O}_1 \cdots \mathcal{O}_n \rangle \langle \tilde{\mathcal{O}}_1 \cdots \tilde{\mathcal{O}}_m \rangle + \mathcal{O}\left(\frac{1}{N}\right), \tag{10}$$

for any $n$ and $m$, one may write

$$\frac{1}{Z^{(II)}} \frac{d^{n+m} Z^{(II)}}{dJ^n d\tilde{J}^m}\bigg|_{J=\tilde{J}=0} = \frac{1}{Z^{(I)}} \frac{d^n Z^{(I)}}{dJ^n}\bigg|_{J=0} \times \frac{1}{Z^{(III)}} \frac{d^m Z^{(III)}}{d\tilde{J}^m}\bigg|_{\tilde{J}=0} + \mathcal{O}\left(\frac{1}{N}\right).$$

Therefore for generalized free field and for large $N$, at leading order, one finds the following relation between partition functions associated with different regions

$$Z^{(II)} \propto Z^{(I)} Z^{(III)}. \tag{11}$$

Actually this equation is a direct consequence of the equation (5) indicating that the physical degrees of freedom in the region $II$ are constructed from those in two other regions. Note that for the case we are considering (eternal black hole) in which the operators $\mathcal{O}$ and $\tilde{O}$ are identical and commuting one finds

$$Z^{(II)} \propto (Z^{(I)})^2. \tag{12}$$

On the other hand using the fact that the partition function may be thought of as the effective action of the vacuum expectation value of the corresponding field when the source is set to zero [20], the above relation may be recast into the following form

$$e^{i\Gamma^{(II)}[\varphi^{(II)}]} \propto e^{2i\Gamma^{(I)}[\varphi^{(I)}]}, \tag{13}$$

where $\varphi = \langle \phi \rangle$ and $\Gamma$ is the effective action. This is, indeed, an equation making a connection between physics describing inside and out side the horizon.

Although we have found the equation (13) for a particular operator, in what follows we would like to consider the effective action for the graviton in which the classical action is given by Einstein-Hilbert action (1). In this case we will linearize the action around the classical solution (2) that may be thought of as the expectation value of the graviton field. Moreover, since at the leading order the effective action is given by the classical action, we should essentially compute on shell action for the solution (2) in different regions.

It is, however, worth noting that the equation (13) contains a proportionality constant which should be fixed if one wants to extract concrete information. Actually, since our ultimate aim is to compute the corresponding effective action (partition function) for the case in which the theory is put at a finite cutoff [3] one may compare two different cases in which the cutoff

---

[3] Our motivation to consider the theory at finite a cutoff comes from $T\bar{T}$ deformation of conformal field theories [21–23] in which it was proposed that the corresponding holographic dual may be provided by gravitational theories with a finite bulk radial cutoff [24–26]. We note, however, that there is a subtlety with Dirichlet boundary condition when we are dealing with the gravity with a finite radial cutoff [27]. Therefore one should be careful when one wants to consider gravity with finite cutoff as a gravitational description for a $T\bar{T}$ deformation. It is worth mentioning that there is another holographic description for the $T\bar{T}$ deformation in terms of the mixed boundary condition [28]. It seems that the $T\bar{T}$ deformation may be holographically better understood in this approach.

Although our main motivation comes from $T\bar{T}$ deformation, our aim was to study the system at finite cutoff which could set by hand. This might happen for example when we have an end of the world brane.

is or is not set to zero.[4] Therefore, we are led to the following equation

$$e^{i(S_{\text{cutoff}}^{(II)} - S_0^{(II)})} = e^{2i(S_{\text{cutoff}}^{(I)} - S_0^{(I)})}, \tag{14}$$

where the proportionality constant is dropped.[5] Here $S^{(II)}$ and $S^{(I)}$ are on shell actions evaluated on the regions $II$ and $I$, respectively.

To proceed, let us compute on shell action for the regions $II$ and $I$ when the cutoff is set to zero. To do so, we note that the action of interest consists of several parts given by

$$S = S_{\text{EH}} + S_{\text{GH}} + S_{\text{CT}}, \tag{15}$$

where $S_{\text{EH}}$ is the Hilbert-Einstein term given by the equation (1) and

$$S_{\text{GH}} = \frac{1}{8\pi G} \int d^{d+1}x \sqrt{-h}K, \qquad S_{\text{CT}} = \frac{-1}{8\pi G} \int d^{d+1}x \sqrt{-h}\frac{d}{L}, \tag{16}$$

are Gibbons-Hawking and counter terms required to have a consistent action with a well defined variation principle that results in a finite free energy. It is, then, straightforward to compute on shell action for the solution (2) in the different regions. In particular for the region $I$ one gets

$$
\begin{aligned}
S_{\text{EH}}^{(I)} &= \frac{\tau V_d L^d}{8\pi G} \left( \frac{1}{r_h^{d+1}} - \frac{1}{\epsilon^{d+1}} \right), \\
S_{\text{GH}}^{(I)} &= \frac{\tau V_d L^d (d+1)}{8\pi G} \left( \frac{1}{\epsilon^{d+1}} - \frac{1}{2r_h^{d+1}} \right), \\
S_{\text{CT}}^{(I)} &= -\frac{\tau V_d L^d d}{8\pi G} \left( \frac{1}{\epsilon^{d+1}} - \frac{1}{2r_h^{d+1}} \right),
\end{aligned}
\tag{17}
$$

so that

$$S_0^{(I)} = \frac{\tau V_d L^d}{16\pi G} \frac{1}{r_h^{d+1}}. \tag{18}$$

Here $\tau$ is a cutoff in the time direction. On the other hand for the region $II$ one finds

$$S_{\text{EH}}^{(II)} = \frac{\tau V_d L^d}{8\pi G} \frac{-1}{r_h^{d+1}}, \quad S_{\text{GH}}^{(II)} = \frac{\tau V_d L^d (d+1)}{16\pi G} \frac{1}{r_h^{d+1}}, \tag{19}$$

and the corresponding counter term vanishes. Therefore, one arrives at

$$S_0^{(II)} = \frac{\tau V_d L^d (d-1)}{16\pi G} \frac{1}{r_h^{d+1}}. \tag{20}$$

Now, let us consider the case in which we have a finite radial cutoff at $r = r_c$ that is associated with a UV cutoff in the dual field theory. It is then straightforward to compute the on shell action for this case. Actually as far as the on shell action for the region $I$ is concerned, the bulk part and the Gibbons-Hawking term of the action have the same expressions except that one needs to replace $\epsilon \to r_c$. On the other hand from the explicit expression of the counter term one gets a non-trivial contribution. Indeed, putting all terms together one arrives at

$$S_{\text{cutoff}}^{(I)} = \frac{\tau V_d L^d}{8\pi G} \left( \frac{1-d}{2r_h^{d+1}} + \frac{d}{r_c^{d+1}} \left( 1 - \sqrt{1 - \frac{r_c^{d+1}}{r_h^{d+1}}} \right) \right), \tag{21}$$

---

[4]In the context of AdS/CFT correspondence the partition function at a finite cutoff has been also studied in [29, 30].

[5]Note that changing the parameters (cutoff) keeps the partition function intact and in the saddle point approximation the overall proportionately constant is cutoff independent.

which may be react into the following form

$$S_{\text{cutoff}}^{(I)} = \frac{\tau V_d L^d}{16\pi G} \left( \frac{1}{r_h^{d+1}} + \frac{d}{r_c^{d+1}} \left( 1 - \sqrt{1 - \frac{r_c^{d+1}}{r_h^{d+1}}} \right)^2 \right). \tag{22}$$

Form this expression it should be evident that the on shell evaluated in the region *II* cannot satisfy the equation (14) unless we make a modification for the on shell action of the inside the horizon too. To proceed, we will assume that there is also a finite radial cutoff behind the horizon located at $r_0$ preventing us to approach the singularity. With this assumption and taking into account all terms contributing to the on shell action one finds

$$S_{\text{cutoff}}^{(II)} = \frac{\tau V_d L^d}{8\pi G} \left( \frac{d-1}{2r_h^{d+1}} - \frac{d}{r_0^{d+1}} \left( 1 - \sqrt{\frac{r_0^{d+1}}{r_h^{d+1}} - 1} \right) \right). \tag{23}$$

Now using the on shell actions with and without cutoff and plugging them into the equation (14), one arrives at

$$\frac{1}{r_0^{d+1}} \left( \sqrt{\frac{r_0^{d+1}}{r_h^{d+1}} - 1} - 1 \right) = \frac{1}{r_c^{d+1}} \left( 1 - \sqrt{1 - \frac{r_c^{d+1}}{r_h^{d+1}}} \right)^2, \tag{24}$$

that is exactly the same expression obtained in [8] relating the cutoff behind the horizon to the UV cutoff. In particular for a small radial cutoff ($r_c \ll r_h$) and at leading order the above equation reduces to

$$r_0 r_c^2 \approx 2^{\frac{4}{d+1}} r_h^3. \tag{25}$$

Although it was not clear from the complexity computations (see [8]) that why the cutoffs $r_0$ and $r_c$ should come with different powers, it should now be evident from the present consideration that it has to do with the fact that operators describing behind the horizon are constructed out of two copies of those describing outside the horizon.

## 4 Discussions

In this paper we have argued that the partition function of the operators describing the interior of an eternal black hole is proportional to the product of partition functions of operators describing left and right exteriors of the black hole. Indeed, this is a direct consequence of the construction of interior operators in terms of those describing the exterior regions proposed in [13]. At leading order this connection may be reduced to a relation between on shell actions evaluated on the inside and outside the black hole.

Using this relation we have computed on shell action for the solution (2) in different regions with the assumption that there is a finite UV cutoff. We have observed that setting a finite UV cutoff enforces us to have a cutoff behind the horizon whose value is fixed by the UV cutoff. Interestingly enough, the expression we have found for the cutoff behind the horizon is the same as that obtained in the context of holographic complexity.

In our computations we have assumed that the UV cutoff is the same for both left and right exterior regions of the corresponding eternal black brane. Nonetheless one could also consider the case in which the UV cutoffs for left and right regions are different. In the context of holographic complexity is was not clear how to proceed in such a situation, though in the present approach it is straightforward to deal with this case. Indeed, the only modification

one needs to make is to compute the on shell action of the two sides with different cutoffs. Doing so, one arrives at

$$\frac{1}{r_0^{d+1}}\left(\sqrt{\frac{r_0^{d+1}}{r_h^{d+1}}-1}-1\right)=\frac{1}{2r_{c_L}^{d+1}}\left(1-\sqrt{1-\frac{r_{c_L}^{d+1}}{r_h^{d+1}}}\right)^2+\frac{1}{2r_{c_R}^{d+1}}\left(1-\sqrt{1-\frac{r_{c_R}^{d+1}}{r_h^{d+1}}}\right)^2, \quad (26)$$

where $r_{c_L}$ and $r_{c_R}$ are radial finite cutoffs associated with the left and the right asymptotic regions. Note that in the limit of small cutoffs, $r_{c_L}, r_{c_R} \ll r_h$, one finds

$$r_0 (r_{c_L}^{d+1} + r_{c_R}^{d+1})^{\frac{2}{d+1}} \approx 2^{\frac{6}{d+1}} r_h^3. \quad (27)$$

Of course, one may consider the case in which one of the cutoff is approaching zero while the other one is kept finite. In this case one could still get a cutoff behind the horizon though its value at leading order is grater than the previous one by a factor of $2^{\frac{2}{d+1}}$.

This situation might be thought of as the case in which the corresponding solution represents a typical black hole microstates of which the gravitational dual is provided by an eternal black hole when only a portion of the left asymptotic region is taken into account [31]. Therefore we are left with a CFT at the boundary of the right exterior side and the left side is capped by a cutoff.

In this case the operators describing the interior of the black hole are still constructed out of two copies of the exterior modes. The second copy associated with the left side is given by the mirror operators [13]. In this case one gets

$$Z^{\text{interior}} \propto Z^{\text{exterior}} Z^{\text{mirror}}, \quad (28)$$

leading to the following expression for the behind the horizon cutoff

$$\frac{1}{r_0^{d+1}}\left(\sqrt{\frac{r_0^{d+1}}{r_h^{d+1}}-1}-1\right)=\frac{1}{2r_{c_L}^{d+1}}\left(1-\sqrt{1-\frac{r_{c_L}^{d+1}}{r_h^{d+1}}}\right)^2, \quad (29)$$

which for small cutoff and at leading order one gets $r_0 r_{c_L}^2 \approx 2^{\frac{6}{d+1}} r_h^3$.

It would be interesting to explore the role of the behind the horizon cutoff and its possible effects in other physical quantities.

## Acknowledgements

We would like to thank A. Mollabashi and K. Papadodimas for useful discussions. M. A. would also like to thank ICTP for very warm hospitality.

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
