# Peer review of "An Infalling Observer and Behind the Horizon Cutoff"

_SciPost Physics, doi:SciPost Phys. 7, 073 (2019)_

## Round 1 · Referee Report · Anonymous · 2019-10-28

Strengths
1- Clear and easy to read
2-Simple example to study the physics in which they are interested
3-Conclusions make sense given assumptions
Weaknesses
1- Not clear how meaningful gravity theory in AdS with finite cutoff is (see report) and does not discuss issues with finite cutoff theories and proposals to get around these issues.
Report
This paper considers the implications of defining AdS gravity with a hard cutoff at large-radius to physics behind the horizon. The motivation stems in principle from the $T\bar{T}$ deformation which was originally stated to be dual to an AdS theory with finite radial cutoff. The authors conclude, by studying generalized free fields in the bulk, that this would imply a finite radial cutoff inside the horizon, by consistency.
The authors make a strong case for why the assumption of a radial cutoff at infinity implies a cutoff behind the horizon. I am however a bit wary of defining quantum gravity in a theory with Dirichlet conditions at finite radial cutoff, in particular because because the classical and quantum perturbation theory is not well defined (see e.g. 1805.11559 for a discussion) as there are linearized instabilities. Furthermore, it has recently been argued (see 1906.11251) that $T\bar{T}$ may be better understood as a theory with mixed boundary conditions at infinity.
I am fully aware that a large number of papers are being published on the subject of these Dirichlet conditions in AdS. I feel strongly, however, that each paper should have a discussion on the conceptual issues with such theories.
Requested changes
1- Discussion of the interest in finite cutoff AdS theories and their physical meaning would be beneficial to the paper.
2- (minor) An explanation around equation (14) as to why the proportionality coefficient should be independent of the cutoff. This should not alter the final conclusion, as far as I understand.
Anonymous on 2019-10-30 [id 637]
We would like to thank the referee for his/her comments. Following the comments one should say:
Actually we agree with the referee that there is a subtlety with Dirichlet boundary condition at finite cutoff as pointed out by Witten. We are also aware of the possible interpretation of TT-bar deformation in terms of mixed boundary condition. It seems to us that the TT-bar deformation may be holographically better understood in this approach. Actually one can also show that working in this description of TT-bar deformation, there is no need to have a cutoff behind the horizon. Nonetheless we should say that although our main motivation comes from TT-bar deformation, our aim was to study the system at finite cutoff which could set by hand. This might happen for example when we have an end of the world brane. In the context of holographic complexity even for the case in which the cutoffs eventually sending to infinite, the behind horizon cutoff might be needed to recover the expected result e.g. such as that one finds for AdS2 case. In any case we think the referee's comment is relevant and raising an interesting issue and therefore we will add a comment on this in the paper with proper citations to the relevance papers.
The referee is right, this point mush be clarified and therefore we will add a comment on this point in the revised version.

---

## Round 2 · Author Response

Following the referee's comments we have revised our paper.

---

## Round 2 · List of Changes

To address the comments, two footnotes are added. footnote 3 and 5. Two papers are also added in the list of references. Ref[27] and Ref[28].

---

## Editorial Decision

published